# Seroprevalence of SARS-CoV-2 and Hepatitis B Virus Coinfections among Ethiopians with Acute Leukemia

**DOI:** 10.3390/cancers16081606

**Published:** 2024-04-22

**Authors:** Jemal Alemu, Balako Gumi, Aster Tsegaye, Ziyada Rahimeto, Dessalegn Fentahun, Fozia Ibrahim, Abdulaziz Abubeker, Amha Gebremedhin, Tesfaye Gelanew, Rawleigh Howe

**Affiliations:** 1Department of Medical Laboratory Sciences, College of Health Sciences, Addis Ababa University, Addis Ababa P.O. Box 1176, Ethiopia; tsegayeaster@yahoo.com; 2Armauer Hansen Research Institute, Addis Ababa P.O. Box 1005, Ethiopia; rahamtoziyada@gmail.com (Z.R.); dessalegnfemu@gmail.com (D.F.); fozia.ibrahim@ahri.gov.et (F.I.); tesfaye.gelanew@ahri.gov.et (T.G.); rawcraig@yahoo.com (R.H.); 3Aklilu Lemma Institute of Pathobiology, Addis Ababa University, Addis Ababa P.O. Box 1176, Ethiopia; balako.gumi@aau.edu.et; 4Department of Internal Medicine, College of Health Sciences, Addis Ababa University, Addis Ababa P.O. Box 1176, Ethiopia; abdulazizas88@gmail.com (A.A.); amhagbr@gmail.com (A.G.)

**Keywords:** SARS-CoV-2, blood-borne viruses, acute leukemia, HBV, Ethiopia

## Abstract

**Simple Summary:**

Immunocompromised individuals, including hematological cancer cases, are prone to a high risk of infections such as SARS-CoV-2. Adverse outcomes, including high mortality, were reported in cases infected with SARS-CoV-2 compared with the general population as well as with non-infected malignancy cases. Liver impairment was reported in individuals co-infected with SARS-CoV-2 and other blood borne viruses including HBV. The aim of this study was to assess the SARS-CoV-2 seroprevalence and occurrence of associated infection with HBV among acute leukemia cases in Ethiopia. The findings of this study could potentially increase the clinician awareness of ongoing infections, improve patient care, and support public health strategies.

**Abstract:**

SARS-CoV-2 and blood-borne viral coinfections are well reported. Nevertheless, little is known regarding the seroprevalence of SARS-CoV-2 and coinfection with blood-borne viruses in hematologic malignancy patients in Ethiopia. We aimed to assess the seroprevalence of SARS-CoV-2 and associated infections with hepatitis B and other viruses among adolescent and adult acute leukemia patients at Tikur Anbessa Specialized Hospital, Addis Ababa, Ethiopia. A cross-sectional study was conducted from July 2020 to June 2021. Blood samples were tested for the presence of anti-SARS-CoV-2, HBV, HCV, and HIV with ELISA kits and occult hepatitis B infection with a real-time polymerase chain reaction assay. Out of a total 110 cases, the SARS-CoV-2 seroprevalence was 35.5%. The prevalence showed a significant increment from July 2020 to the end of June 2021 (*p* = 0.015). In 22.7% and 2.7% of leukemia cases, HBV and HIV, respectively, were detected. No HCV was identified. The rate of SARS-CoV-2 coinfection with HBV and HIV was 28% (11/39) and 2.6% (1/39), respectively; however, there was no statistically significant association between SARS-CoV-2 seropositivity with HBV and HIV (*p* > 0.05). There is a need for viral screening in leukemia cases to monitor infections and inform management.

## 1. Introduction

COVID-19 is a disease caused by severe acute respiratory syndrome coronavirus 2 (SARS-CoV-2) [1] that has spread rapidly throughout the world since its emergence in December 2019 and affects millions of people [2]. The symptoms range from mild fever to severe, including difficulties in breathing, multiorgan failure, and death [3]. Patients with neoplastic diseases who are infected by SARS-CoV-2 are at an increased risk for various complications including the need for intensive care and death relative to non-cancer cases [4]. A study revealed a high COVID-19 mortality rate in patients with cancer [5], including hematological malignancies [6]. COVID-19 impacts leukemia patients in many ways, including missed or delayed diagnosis, delay or deferral of chemotherapy and/or hematopoietic stem cell transplantation, shortages of transfusion blood products, or the interruption of maintenance therapy [7].

Hepatitis B virus causes chronic infection and is one of the main health problems in the world; it is particularly high in low-income countries, such as countries in Africa and Asia [8]. It causes liver cell injury due to the immune response elicited by the infected hepatic cells and causes more than one million deaths each year due to liver cancer and cirrhosis [9]. In leukemia patients, asymptomatic or subclinical coinfection of SARS-CoV-2 with hepatitis viruses such as HBV may target the liver, resulting in poor prognosis [10] since both viruses can damage liver cells/tissues [11]. Moreover, the liver function of patients infected with SARS-CoV-2 is highly affected due to the presence of HBV coinfection compared to SARS-CoV-2-infected cases without HBV [12].

Assessing the prevalence of viral pathogens can increase clinician awareness, lead to additional laboratory test-based healthcare service to patients, enhance leukemia patients’ care, and support public health strategies [13,14].

Ethiopia reported its first COVID-19 case on 13 March 2020 [15]. As of 13 December 2023, more than 501,117 cases and 7574 deaths had been reported [16]. Several studies have been conducted in developed countries on the burden of COVID-19 in solid tumors and hematologic malignancies [17,18]. Screening of leukemia patients for viruses, including SARS-CoV-2 and HBV, is critical for early diagnosis and management of the disease by preventing delay or deferral in chemotherapy and stem cell transplantation and decreasing the severity as well as the complications [19].

The seroprevalence of SARS-CoV-2 and associated infection with HBV among hematology malignant cases is not well documented in resource-limited countries. This present study evaluated these issues, focusing on acute leukemia patients attending at Tikur Anbessa Specialized Hospital (TASH), Addis Ababa, Ethiopia. The present study was the first on leukemia cases. Highlighting the importance of laboratory screening for the presence of viral pathogens in future studies with large sample sizes.

## 2. Materials and Methods

### 2.1. Study Design and Participants

A cross-sectional study was conducted from July 2020 to June 2021. A total of 110 SARS-CoV-2 unvaccinated adult and adolescent (13–17 years old) acute leukemia patients who consecutively visited the hematology clinic of TASH, Addis Ababa, Ethiopia, were recruited; patients had not received chemotherapy.

### 2.2. Data and Sample Collection

A standardized data collection form was used to collect sociodemographic as well as clinical characteristics, 3 mL measures of blood samples were collected (at one time) in EDTA tubes, and, subsequently, plasma was extracted and stored at −70 °C until laboratory investigation. Plasma samples collected from sixty-six patients at TASH who had been diagnosed with acute leukemia prior to the initial COVID-19 report in Ethiopia were included as a presumptive control population negative for SARS-CoV-2 exposure.

### 2.3. SARS-CoV-2 Seroprevalence Testing

Plasma samples were screened for the presence of anti-SARS-CoV-2 receptor binding domain (RBD) of spike protein IgG antibodies using a validated in-house enzyme-linked immunosorbent assay (ELISA) as described [20]. Positive samples were confirmed with the SARS-CoV-2 Wantai Ab ELISA, Beijing Biological Pharmacy Enterprise Co., Ltd., Beijing, China, which is a commercially available kit used to detect the total antibodies (IgG, IgM and IgA) against S-RBD, at the immunology laboratory of the Armauer Hansen Research Institute (AHRI). Recruited patients had not been tested using a PCR for COVID-19 disease at the time of sample collection.

### 2.4. HBV Laboratory Testing

Plasma samples were screened using an HBsAg enzyme immunoassay (HBsAg ELISA-Beijing Wantai Biological Pharmacy Enterprise Co., Ltd., Beijing, China), and negative samples were tested for both IgM and IgG using anti-HBcAg ELISA kits (Monolisa Anti-HBc PLUS, Bio-Rad, Marnes-la-Coquette, France). The plasma samples that tested positive for anti-HBcAg were further analyzed using PCR for HBV DNA [21].

Briefly, 200 µL of each plasma sample was subjected to DNA extraction, amplification, and detection at the molecular biology laboratory of ALERT hospital using a commercially available real-time PCR platform (Abbott Molecular Inc., 1300 East Touhy Avenue, Des Plaines, IL 60018 USA) with an Abbott m2000rt instrument, with a lower detection limit of <1.18 log IU/mL genome equivalent to 15 IU/mL, to determine occult hepatitis B infection (OBI). Microbiologically detectable HBV infection was defined as either HBsAg positive or HBsAg negative but HBV DNA positive [21].

### 2.5. HIV Laboratory Testing

An enzyme-linked immunosorbent assay (Micro ELISA-HIV Ag & Ab, J. Mitra & Co. Pvt. Ltd., New Delhi, India) was used to screen for HIV to detect HIV-1 and/or HIV-2 and HIV-1 p24 antigen. Rapid tests using the Ethiopian national algorithm were again performed on positive samples: SD HIV 1/2 3.0, Standard Diagnostics, Inc. (Kyonggi-do, South Korea); ABON, Abon Biopharm Co., Ltd., (Hangzhou, China); and CHEMBIO, Chembio Diagnostic Systems, Inc. (Medford, NY, USA) [21].

### 2.6. HCV Laboratory Testing

Anti-HCV ELISA, Beijing Wantai Biological Pharmacy Enterprise Co., Ltd., Beijing, China, was used to screen for HCV. Positive samples were then examined using the Wondfo one-step fast antibody test for HCV, Guangzhou Wondfo Biotech Co., Ltd., Guangzhou, China [21].

### 2.7. Quality Assurance

Every laboratory test including quality control was carried out in compliance with the manufacturer’s guidelines. Every quality control step was carried out in accordance with standard operating procedures. ELISA tests were executed by running the samples twice and calculating the average. The PCR test was run with positive and negative controls.

### 2.8. Data Analysis

The collected data were coded and entered on the computer using SPSS version 25 statistical software and analyzed with descriptive statistics, including the mean, median, range, standard deviation, and percentage. The included variables were age, sex, residential area, leukemia subtype, sample collection date, HBV, HCV, and HIV prevalence. The association between variables was analyzed using the chi-square test. A *p*-value of <0.05 was considered statistically significant.

### 2.9. Ethical Clearance

The present study was approved by the AHRI/ALERT Ethics Review Committee (AAERC) on 19 November 2018, with Protocol No. PO34/18, with an amendment as a waiver of patient consent approved for viral serology.

## 3. Results

### 3.1. Sociodemographic and Clinical Characteristics of Study Participants

Out of a total 110 acute leukemia cases, 64 (58.2%) were male. The median age was 25, with an IQR of 19–37 and a range of 13–76 years. The majority of the cases were in the age range of 18–39, with the fewest cases in the age group of 60 and above. The distribution of leukemia included 52 (47.3%) cases of acute lymphocytic leukemia (ALL), 44 (40%) cases of acute myelogenous leukemia (AML), 12 (10.9%) cases of acute leukemia not specified, and 2 cases (1.8%) of chronic myelogenous leukemia presenting in acute blast crisis. Most patients, 61.8% (68/97), presented with anemia. Only 9% (9/99) of the patients resided in Addis Ababa, the site of the hospital study; the remainder, 91% (90/99), came from diverse regions in Ethiopia outside of Addis Ababa (Table 1).

### 3.2. Seroprevalence of SARS-CoV-2 and Its Distribution

The SARS-CoV-2 seroprevalence was assessed among 110 cases enrolled after the presumed first case of COVID-19 in Ethiopia, and the seroprevalence among all cases was found to be 35.5% (39/110). Seropositivity did not differ between males (36%) and females (35%). The age group of 18–39 years (40%, 27/68) did not differ from that of 40–59 years (38%, 6/16). In contrast, a much lower prevalence (14%, 1/7) was observed in the age group of 60 and older. The seropositivity of the virus among ALL and AML cases was 31% and 39%, respectively. Meanwhile, among the 110 cases, there was no statistically significant association between SARS-CoV-2 seropositivity and sex, age category, residency area, and type of leukemia diagnosis (Table 1).

Figure 1 summarizes the seropositivity of samples according to the date of collection, stratified into five time periods, one prior to March 2020 and four periods each of 3 months in duration from July 2020 to June 2021. Seropositivity was only detectable subsequent to July 2020 and thereafter increased steadily from 15% to 56% during the final time period (April–June 2021) of sample collection. There was a statistically significant difference in the seropositivity of samples collected during the five different time periods (*p* = 0.015, chi-square). Importantly, we evaluated 66 leukemia cases whose plasma samples were obtained prior the initial March 2020 case of SARS-CoV-2 in Ethiopia, and these cases were all negative, confirming the specificity of the in-house anti-RBD IgG detection ELISA. Moreover, a random subset of these pre-COVID-19 cases was confirmed to be seronegative using commercially available kits.

The distribution of SARS-CoV-2 seropositivity in different regions of the country is depicted in Figure 2. SARS-CoV-2 seroprevalence cases were observed among acute leukemia patients who came from six regions, namely Oromia, Amhara, Tigray, Southern Ethiopia, Somalia, and Addis Ababa, but not from Harari or Afar, though only one and two patients were from the latter regions, respectively. Chi-square testing did not reveal a statistically significant difference in SARS-CoV-2 seroprevalence across all regions (*p* = 0.83). SARS-CoV-2 seropositive samples from Addis Ababa were observed after August 2020, from the Oromia, Amhara, and Southern Ethiopia regions after September 2020, from the Somalia region after January 2021, and from Tigray after April 2021. The association of SARS-CoV-2 seropositivity with the time and region variables was also assessed in a binary logistic regression multivariate model, which confirmed the association with the time of sample collection (*p* = 0.010) and not region (*p* = 0.226).

### 3.3. Seroprevalence of SARS-CoV-2 in Relation to Other Studies in Ethiopia

As shown in Figure 3, we compared our results with multiple other studies of non-cancer subjects performed in Ethiopia over the same time period. Six of these studies recruited healthcare workers and associates; two included hospitalized patients [22,23]. Positivity was reported from late April 2020 to September 2020, with a range of 1.9% to 22.9% in different urban and rural areas of Ethiopia [22,23,24,25], whereas, among frontline health workers and communities, prevalences of 25% and 35% in November–December 2020 and February–March 2021, respectively, were reported [26]. Moreover, a prevalence of 39.6% was reported from December 2020 to February 2021 among health workers in different regions of the country and 51.8% in school children in Hawassa [20,27]. As can be observed in Figure 3, when our results are juxtaposed with these studies according to time period, there do not appear to be any striking differences in seroprevalences between our study and previous studies, considering the overall increasing trend in cases. Notably, the prevalence observed in our leukemia cohort was very similar to that observed by Gelanew et al. [20], performed by our institute using the same serological assay but focusing on healthcare providers in multiple regions of the country during similar time frames.

### 3.4. Blood-Borne Virus Distribution and Association with SARS-CoV-2 Seropositivity

We also tested plasma samples from leukemia patients for other viruses: HBV, hepatitis C virus (HCV), and human immunodeficiency virus (HIV) [21]. Out of a total of 110 cases, the prevalence of HBV infection was 22.7% (25/110). This represents 12.7% (14/110) positive for HBsAg and a further 10.0% HBsAg negative but HBV DNA positive. Females were more often infected with active HBV infection (10 females versus 4 males), while males were more often infected with occult HBV infection (10 males versus 1 female); however, there was no statistically significant association between total HBV infection and gender (*p* > 0.05). The HIV seropositivity was 2.7%, and no HCV was identified (Table 2). In addition, of the 66 leukemia cases identified prior to the COVID-19 pandemic, the percentage positive for HBsAg was 7.6% (5/66); the percentage negative for HBsAg but with detectable HBV DNA was 12.6% (8/66). Among the pre-COVID-19 leukemia cases, 4.5% of them were positive for HCV (3/66), and none were positive for HIV.

Among SARS-CoV-2 seropositive cases, associated infection with HBV was observed in 11/39 (28%), and associated infection with HIV was observed in 1/39 (2.6%) of those cases; however, there was no statistically significant association between SARS-CoV-2 seropositivity and HBV or HIV (*p* > 0.05) (Table 2). Moreover, there was no statistically significant association between the leukemia subtype and HBV infection status among SARS-CoV-2 seropositive patients (*p* > 0.05) (Table 3).

Moreover, the effect of seropositivity to SARS-CoV-2 on liver function assessed with ALT (alanine aminotransferase) revealed a median value of 15 IU/mL [IQR, 14–44] among those with an associated HBV infection and a median value of 15 IU/mL [IQR, 9.7–19] among SARS-CoV-2 seronegative cases infected with HBV. Moreover, 90% and 85% of SARS-CoV-2 with associated HBV and HBV mono-infected SARAS-Cov-2 seronegative cases, respectively, had normal ALT levels; these differences were not statistically significant (*p* > 0.05) (Table 4).

## 4. Discussion

Viral infections, including that of SARS-CoV-2, complicate cancer management [29]; indeed, enhanced mortality risk among SARS-CoV-2-infected cancer patients has been reported [30]. Immune system impairment, including T cell exhaustion due to cancer, as well as immune suppressive chemotherapy, contribute to suboptimal immune responses and worse outcomes in COVID-19 disease [31]. Moreover, subtypes of leukemia also differ in severity and outcomes of COVID-19 due to underlying differences in disease pathogenesis and different chemotherapy regiments [32].

In the present study, the SARS-CoV-2 seroprevalence in acute leukemia patients was 35.5%. There was a statistically significant difference in the seropositivity of samples collected at different time periods. SARS-CoV-2 seropositivity was reported in the majority of regions of Ethiopia, and the prevalence of SARS-CoV-2 seropositivity was consistent with previous studies conducted in Ethiopia. In addition, 22.7% of the leukemia cases were positive for HBV, and 28% of SARS-CoV-2-seropositive cases were found to be HBV infected. However, there were no statistically significant associations between SARS-CoV-2 seropositivity and HBV, HCV, or HIV seropositivity, or with sex, age category, Ethiopian region of residence, or acute leukemia subtypes. There was no difference in the levels of the liver marker ALT among HBV-infected SARS-CoV-2 seropositive and seronegative cases. These findings may reflect the relatively small sample size of the study or inadequate antibody responses to SARS-CoV-2 due to being immunocompromised.

In line with this study, seropositivity was reported in Italy and the United Kingdom among leukemia cases [33,34] and among oncology cases in South Africa [35]. The present study showed that the seroprevalence of SARS-CoV-2 was distributed in several parts of the country with different degrees of burden. Within Ethiopia, population-based household surveillance conducted in Addis Ababa and Jimma from 22 July 2020 to 2 September 2020 reported that there was 0.5–1.9% positivity for IgG antibodies to the infection [28].

Conversely, a SARS-CoV-2 seroprevalence study conducted from December 2020 to February 2021 in Ethiopia among health workers from public hospitals working in different regions of the country reported that the overall SARS-CoV-2 seroprevalence was 39.6% [20]. Furthermore, a systematic review in Ethiopia revealed that the pooled prevalence of SARS-CoV-2 infection was 8.8% [36].

Our study finding of a SARS-CoV-2 prevalence of 35.5% may be considered high in comparison to other studies in Ethiopia. However, it is important to place the time of the studies in context. Many of the previous studies in Ethiopia were carried out in the early stages of the COVID-19 pandemic [24,25]. Certainly, the numbers of seropositive cases increased as the epidemic progressed, and our period of sampling covered that in which new variants emerged in Ethiopia and elsewhere [35,37]. According to national data, throughout all periods of the study, strains B.1/B.1.1 and B.1.480 were seen. The alpha strain appeared in early 2021, and the delta variant emerged in spring of 2021 [37]. However, we did not type these strains in our study, so we do not know of the impact of strain diversity on our observed prevalence. In contrast to another study on hematological malignancies, our study did not find significant differences in the SARS-CoV-2 antibody quantity as defined by plasma optical density [38].

SARS-CoV-2 seroprevalence data may be influenced by factors unique to patients with malignancies, such as different rates of exposure perhaps related to a more frequent presence in health facilities for leukemia patients [38] or differences in seroconversion rates related to underlying immunosuppression. These factors are difficult to assess. The simplest interpretation is that the SARS-CoV-2 seroprevalence in this leukemia population reflects national trends.

The present study assessed not only SARS-CoV-2 seroprevalence but also the burden of HBV and other viral infections in those patients. Since the HBV prevalence in the country is high, the effect of SARS-CoV-2 on HBV-infected cases potentially impacts patient management [39]. SARS-CoV-2 uses the angiotensin-converting enzyme 2 (ACE-2) receptor, which is found in several organs, including the liver, and is highly expressed in bile duct cells and in pre-existing liver disease. It is preferred by the virus, leading to liver cell damage through cholangiocyte dysfunction or inflammation. Furthermore, other molecules, such as dipeptidyl peptidase 4 and transmembrane serine protease 2, found in organs such as the liver, may also be host cell receptors for the virus [40,41].

In the present study, SARS-CoV-2 coinfection with HBV was detected, and this finding was consistent with other study [39]; moreover, liver function was addressed and indicated that exposure to SARS-CoV-2 had no direct effect on the liver as assessed by ALT levels. In contrast, other studies reported that the degree of liver injury was high in COVID-19 patients with HBV coinfection [12]. There may be a need to re-address this issue with larger sample sizes and more liver function tests. None of the clinical variables, including the leukemia patients’ age group, leukemia type, HBV, HIV, and HCV, and liver ALT levels, were found to be significantly associated with SARS-CoV-2 seropositivity. This is consistent with the findings of research on cancer cases in South Africa [35] and research on health professionals in Ethiopia [20].

### 4.1. Strength

This was the first study conducted on leukemia cases in Ethiopia, and the study participants were representative of multiple regions of the country. Moreover, none of the study participants were vaccinated for the virus, indicating that the seroprevalence was directly related to SARS-CoV-2 exposure.

### 4.2. Limitation

No molecular tests were conducted to determine whether any patients had an active SARS-CoV-2 infection; however, our IgG seropositive cases were confirmed with commercially available serological kits and found to be positive for IgG, IgM and IgA. Though the sample size was adequate to provide estimates of SARS-CoV-2 seroprevalence, a larger sample size would have improved the possibility of establishing associations between SARS-CoV-2 infection and other clinical variables. Finally, we included independent studies of the seroprevalence of SARS-CoV-2 among primarily healthcare workers across time and region in Ethiopia to provide a perspective on whether the seroprevalence among the leukemia patients in this study was strikingly similar or different. A better comparator would have been a planned non-cancer cohort otherwise matched with our leukemia patients, but this would be difficult given the diverse regions from which leukemia patients in this referral hospital setting originate.

## 5. Conclusions

Both SARS-CoV-2 and HBV infection were found to be highly prevalent in our leukemia patient cohort, suggesting the need for viral screening of hematological malignant cases to monitor infections and improve prognosis. Future studies with larger sample sizes may aid in delineating the patient outcomes of viral coinfections among leukemia patients.

## Figures and Tables

**Figure 1 cancers-16-01606-f001:**
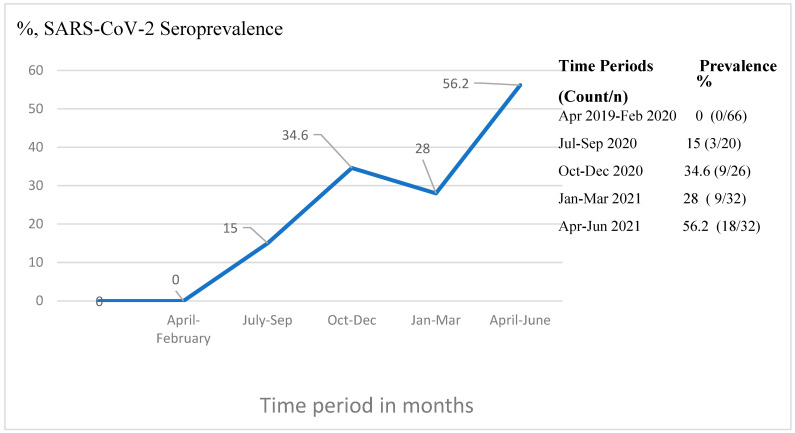
SARS-CoV-2 seroprevalence over time periods among acute leukemia patients admitted to TASH, Addis Ababa, Ethiopia.

**Figure 2 cancers-16-01606-f002:**
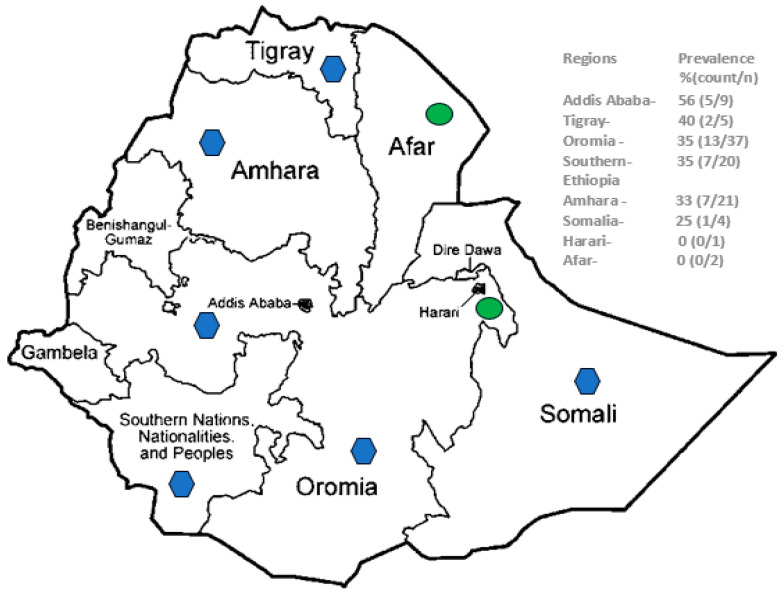
SARS-CoV-2 seroprevalence among acute leukemia patients admitted to TASH, Addis Ababa from different regions of Ethiopia. Key: blue hexagon—regions with SARS-CoV-2 seropositive leukemia patients; green circle—regions with SARS-CoV-2 seronegative leukemia patients.

**Figure 3 cancers-16-01606-f003:**
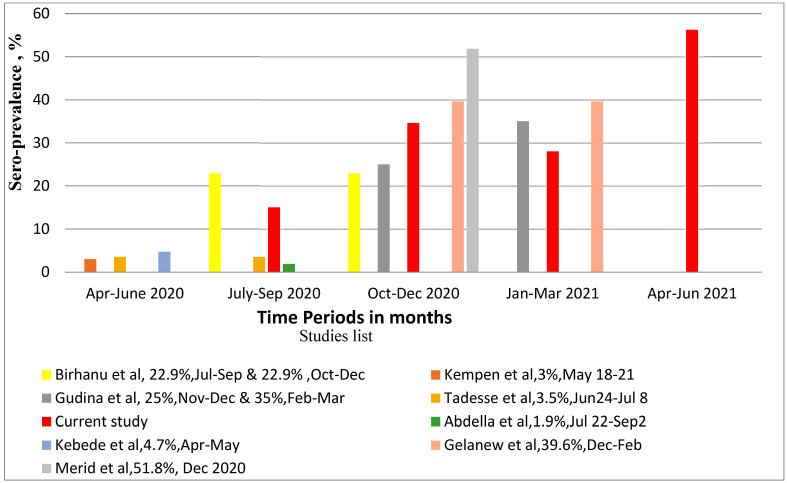
SARS-CoV-2 seroprevalence report among studies conducted in Ethiopia [20,22,23,24,25,26,27,28].

**Table 1 cancers-16-01606-t001:** Baseline demographic, clinical characteristics and associated factors of SARS-CoV-2 seroprevalence among adolescent and adult acute leukemia patients, n = 110.

Variables	Anti-SARS-CoV-2 RBD-IgG Detection		Total Count (%)	*p* Value *
	Positive, n (%)	Negative, n (%)		
**Sex**				0.901
Male	23 (36)	41 (64)	64 (58)	
Female	16 (35)	30 (65)	46 (42)	
**Age group (in years)**				0.452
13–17	5 (26)	14 (74)	19 (17.3)	
18–39	27 (40)	41 (60)	68 (61.8)	
40–59	6 (38)	10 (62)	16 (14.5)	
60 and above	1 (14)	6 (86)	7 (6.4)	
**Residency**				0.833
Addis Ababa	5 (56)	4 (44)	9 (9)	
Other regions in Ethiopia	30 (33)	60 (67)	90 (91)	
**Leukemia Diagnosis**				0.407
ALL	16 (31)	36 (69)	52 (47.3)	
AML	17 (39)	27 (61)	44 (40)	
CML with blast crisis	0 (0)	2 (100)	2 (1.8)	
Acute leukemia, not specified	6 (50)	6 (50)	12 (10.9)	

* Chi-square result; *p* value < 0.05 indicates statistical significance; ALL—acute lymphocytic leukemia; AML—acute myelogenous leukemia; CML—chronic myelogenous leukemia; RBD—receptor binding domain.

**Table 2 cancers-16-01606-t002:** Association of SARS-CoV-2 seroprevalence with blood-borne viruses among acute leukemia patients attended at TASH, Addis Ababa Ethiopia.

Blood-Borne Viral Infections	Viral Markers	Viral Marker Detection	Anti-SARS-CoV-2 RBD-IgG Detection		Total Count (%)	*p* Value *
			Positive, n (%)	Negative, n (%)		
**HBV**	HBsAg	Positive	7 (50)	7 (50)	14 (12.7)	0.223
		Negative	32 (33)	64 (67)	96 (87.2)	
	HBV-DNA	Positive	4 (36)	7 (64)	11 (10)	0.947
		Negative	35 (35)	64 (65)	99 (90)	
	Total HBV	Positive	11 (44)	14 (56)	25 (22.7)	0.310
	(HBsAg + HBV-DNA)	Negative	28 (33)	57 (67)	85 (77.3)	
**HIV**	Anti-HIV	Positive	1 (33)	2 (67)	3 (2.7)	0.938
		Negative	38 (36)	69 (64)	107 (97.3)	
**HCV**	Anti-HCV	Positive	0	0	0 (0)	---
		Negative	39 (35)	71 (65)	110 (100)	

* Chi-square result; *p* value < 0.05 indicates statistical significance; HBsAg—hepatitis B surface antigen; HBV—hepatitis B Virus; HIV—human immune-deficiency virus; HCV—hepatitis C virus; RBD—receptor binding domain.

**Table 3 cancers-16-01606-t003:** Leukemia diagnosis types among SARS-CoV-2 seropositive cases.

Leukemia Diagnosis	HBV Infection Status		Total	* *p* Value
	Positive	Negative		
ALL	4	14	11	0.2273
AML	7	10	28	
Total	11	24	35	

* Chi-square result; *p* value < 0.05 indicates statistical significance; ALL: acute lymphocytic leukemia. AML: acute myelogenous leukemia. HBV: hepatitis B Virus.

**Table 4 cancers-16-01606-t004:** Effect of SARS-CoV-2 serostatus and HBV infection on ALT.

Number of Cases with Indicated Serostatus	ALT Level		Total	* *p* Value
	0–45 IU/mL	≥46 IU/mL		
SARS-CoV-2 Pos, HBV pos	10	1	11	0.6624
SARS-CoV-2 Neg, HBV pos	24	4	28	
Total	34	5	39	

* Chi-square result; *p* value < 0.05 indicates statistical significance; HBV—hepatitis B virus; ALT—alanine aminotransferase.

## Data Availability

The datasets analyzed and presented in this study are available from the corresponding authors on reasonable request and are not publicly available due to privacy or ethical restrictions.

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
