# Peer review of "Seroprevalence of SARS-CoV-2 and Hepatitis B Virus Coinfections among Ethiopians with Acute Leukemia"

_cancers, 2024, doi:10.3390/cancers16081606_

Round 1
Reviewer 1 Report
Comments and Suggestions for Authors
Dear Authors,
The manuscript under review “Seroprevalence of SARS-CoV-2 and Hepatitis B virus Coinfections Among Ethiopians with Acute Leukemia”
The clinical parameter of (SARS-CoV-2 + HBV+ acute leukemia (ALL, AML, CML)) will be very interesting to see the cross talk. A total of 110 SARS-Co-2 with acute leukemia patients with clinical record of hematological parameter before chemotherapy, participated before chemotherapy was initiated on them.
1. In over-all, I think the knowledge of this article is interesting and the authors' interesting observations on this topic may be of interest to the readers. However, some comments need to be addressed to improve the quality of the article, its adequacy, and its readability prior to the publication in the present form. My overall judgment is to publish this article after the authors have carefully considered my suggestions below.
2. In my opinion, some recent references for the year 2023-24 are highly recommended. Therefore, I suggest the authors focus their efforts on researching the most recent and relevant literature: I believe that adding a few more studies will help to provide better and more accurate background to this study. Please on the link (https://scholar.google.com/scholar?as_q=Hepatitis+B&as_epq=SARS+CoV+2&as_oq=&as_eq=&as_occt=title&as_sauthors=&as_publication=&as_ylo=2023&as_yhi=&hl=en&as_sdt=0%2C5&as_vis=1
3. As the reviewer desired to see the overall study design before going into detail of manuscript. Hence the author is suggested to incorporate comprehensive flow sheet abstract (colored Scheme), which must be part of the manuscript, including all information that is ranging from material experiment parts, results and other key points mentioned in the text. This part is very important, as the reader will understand the whole manuscript without going into detail. See the following for example: https://doi.org/10.1080/21655979.2020.1865607
4. One page: Similarly infographic abstract should also part of the manuscript, covering all the detail mentioned in the manuscript, especially pictures, where it is required, for ready reference, https://doi.org/10.1080/21655979.2020.1867405
5. Why was this study done? (Separate paragraph)
6. What did the author and co-authors do and find? (Separate paragraph)
7. What do these findings mean? (Separate paragraph)
8. What is the impact of this research on society? (Separate paragraph)
9. Key highlights/ Future direction must also be part of the manuscript. (Separate paragraph)
10. Table 1, Table 2., some typing errors make it difficult to understand it.
It very difficult to find the missing words, please correct the table and resubmit for review.
Comments on the Quality of English LanguageTyping and spell error must be corrected for review
Author Response
Thank you for your valuable comments
please find the document(Revised mauscriprt)
Response to Reviewer II
We appreciate your valuable comments.
We presented our response by highlighting it with a yellow color in the main document.
Very interesting study also regarding epidemiology in patients with leukemia in a geographical area with sometimes incomplete epidemiological data. It would be very interesting to evaluate the phase of the disease in which positivity for SARS-CoV-2 occurred in order to evaluate whether serological responses occurred in patients undergoing chemotherapy or not. The data about the treatment phase (at least for large groups) should be entered as well as if there are pre- and post-infection lymphocyte typifications
Response : We thank the reviewer for raising these important points.
Point by point response for Reviewer I
We would appreciate your constructive comments.
We presented our response by highlighting it with a yellow color in the main document.
- In my opinion, some recent references for the year 2023-24 are highly recommended. Therefore, I suggest the authors focus their efforts on researching the most recent and relevant literature: I believe that adding a few more studies will help to provide better and more accurate backgrounds this study. Please on the link (https://scholar.google. com/scholar? as_q= Hepatitis +B&as_epq=SARS+CoV+2&as_oq=&as_eq=&as_occt=title&as_sauthors=&as_publication=&as_ylo=2023&as_yhi=&hl=en&as_sdt=0%2C5&as_vis=1
Response: We thank the reviewer for the valuable suggestions. In response, in the Introduction section, paragraph 2, we added references 8, 9, 10, 12 and 13, recent papers that you suggested us to add). — see Page 4, from line 93-101.
- incorporate comprehensive flow sheet abstract (colored Scheme), which must be part of the manuscript, including all information that is ranging from material experiment parts, results and other key points mentioned in the text. This part is very important, as the reader will understand the whole manuscript without going into detail. See the following for example: https: //doi.org /10.1080/21655979.2020.1865607
Response: We thank the reviewer for raising this important point. In response, we produced and inserted an abstract along with a flow chart. —see Line 56 of Page 2 of the revised manuscript
- One page: Similarly, infographic abstract should also part of the manuscript
Response: We thank you for such valuable suggestions that strengthened the quality of our manuscript. In response, we produced and inserted an infographic abstract. – see line 76 of page 3 of the revised manuscript.
- Why was this study done?
Response: Thank you for this comment. It was overlooked in the first submission and thanks to you for highlighting now included in the introduction section, paragraph 3 of lines 102-to 105 of the revised manuscript.
- What did the author and co-authors do and find? (Separate paragraph)
Response: This is to kindly inform you that this information is stated in the Materials and Methods section on pages 5, 6, 7, and in the results on pages 8–15. However, we tried to present it in one paragraph in concise form, as presented below.
A cross-sectional study was conducted from July 2020 to June 2021 among acute leukemia cases. A standardized data collection form was used to collect demographic and clinical data, and 3 ml of blood samples were collected. Plasma samples were tested with ELISA for the presence of anti-SARS-CoV-2, HBsAg, anti-HIV, and anti-HCV, and PCR was used for HBV-DNA. The collected data were analyzed using SPSS version 25 statistical software, and the association between variables was analyzed by chi-square. A P-value of <0.05 was considered statistically significant. The SARS-CoV-2 seroprevalence was 35.5%. There was a statistically significant difference in the seropositivity of samples collected at different time periods. SARS-CoV-2 seropositivity was reported in the majority of regions of Ethiopia, and the prevalence of SARS-CoV-2 seropositivity was consistent with previous studies conducted in Ethiopia. In addition, 22.7% of the cases were positive for HBV, and 28% of SARS-CoV-2-seropositive cases were found to be positive for HBV infection. However, there were no statistically significant associations between SARS-CoV-2 seropositivity and HBV, HCV, or HIV seropositivity, or with sex, age category, Ethiopian region of residence, or acute leukemia subtypes. There was no difference in the liver function test or ALT between SARS-CoV-2 seropositive and HBV-infected cases and cases with seronegative SARS-CoV-2 infected with HBV.
- What do these findings mean? (Separate paragraph)
Response: We would like to mention that this information is mentioned in the Conclusion section of the revised manuscript, page 20, lines 508–511. And presented below
Both SARS-CoV-2 and HBV infections were found to be highly prevalent in our leukemia patients’ cohort, suggesting the need for viral screening in hematological malignant cases to monitor infections and improve prognosis. Future studies with larger sample sizes may aid in delineating the patient outcomes of viral coinfections among leukemia patients.
- What is the impact of this research on society
è These points added in the Introduction section on lines 102-105, page 4 and on lines 119-121., page 5
- Key highlights/ Future direction must also be part of the manuscript. (Separate paragraph)
Response: - We agreed with the reviewer's comment. This recommendation
” Although the present study was the first on leukemia cases, highlighting the importance of laboratory screening for the presence of viral pathogens, future studies on the incidence of viral pathogens among leukemia patients at a large sample size is needed’’
is added in the revised manuscript. On lines 122-124, page 5
- Table 1, Table 2., some typing errors make it difficult to understand it.
It very difficult to find the missing words, please correct the table and resubmit for review.
Response: Comments were accepted, and corrections were done accordingly.
We inserted/corrected word/s in Table 1(line 280 and Table 2(line 377) -
Point by point response for Reviewer I
We would appreciate your constructive comments.
We presented our response by highlighting it with a yellow color in the main document.
- In my opinion, some recent references for the year 2023-24 are highly recommended. Therefore, I suggest the authors focus their efforts on researching the most recent and relevant literature: I believe that adding a few more studies will help to provide better and more accurate backgrounds this study. Please on the link (https://scholar.google. com/scholar? as_q= Hepatitis +B&as_epq=SARS+CoV+2&as_oq=&as_eq=&as_occt=title&as_sauthors=&as_publication=&as_ylo=2023&as_yhi=&hl=en&as_sdt=0%2C5&as_vis=1
Response: We thank the reviewer for the valuable suggestions. In response, in the Introduction section, paragraph 2, we added references 8, 9, 10, 12 and 13, recent papers that you suggested us to add). — see Page 4, from line 93-101.
- incorporate comprehensive flow sheet abstract (colored Scheme), which must be part of the manuscript, including all information that is ranging from material experiment parts, results and other key points mentioned in the text. This part is very important, as the reader will understand the whole manuscript without going into detail. See the following for example: https: //doi.org /10.1080/21655979.2020.1865607
Response: We thank the reviewer for raising this important point. In response, we produced and inserted an abstract along with a flow chart. —see Line 56 of Page 2 of the revised manuscript
- One page: Similarly, infographic abstract should also part of the manuscript
Response: We thank you for such valuable suggestions that strengthened the quality of our manuscript. In response, we produced and inserted an infographic abstract. – see line 76 of page 3 of the revised manuscript.
- Why was this study done?
Response: Thank you for this comment. It was overlooked in the first submission and thanks to you for highlighting now included in the introduction section, paragraph 3 of lines 102-to 105 of the revised manuscript.
- What did the author and co-authors do and find? (Separate paragraph)
Response: This is to kindly inform you that this information is stated in the Materials and Methods section on pages 5, 6, 7, and in the results on pages 8–15. However, we tried to present it in one paragraph in concise form, as presented below.
A cross-sectional study was conducted from July 2020 to June 2021 among acute leukemia cases. A standardized data collection form was used to collect demographic and clinical data, and 3 ml of blood samples were collected. Plasma samples were tested with ELISA for the presence of anti-SARS-CoV-2, HBsAg, anti-HIV, and anti-HCV, and PCR was used for HBV-DNA. The collected data were analyzed using SPSS version 25 statistical software, and the association between variables was analyzed by chi-square. A P-value of <0.05 was considered statistically significant. The SARS-CoV-2 seroprevalence was 35.5%. There was a statistically significant difference in the seropositivity of samples collected at different time periods. SARS-CoV-2 seropositivity was reported in the majority of regions of Ethiopia, and the prevalence of SARS-CoV-2 seropositivity was consistent with previous studies conducted in Ethiopia. In addition, 22.7% of the cases were positive for HBV, and 28% of SARS-CoV-2-seropositive cases were found to be positive for HBV infection. However, there were no statistically significant associations between SARS-CoV-2 seropositivity and HBV, HCV, or HIV seropositivity, or with sex, age category, Ethiopian region of residence, or acute leukemia subtypes. There was no difference in the liver function test or ALT between SARS-CoV-2 seropositive and HBV-infected cases and cases with seronegative SARS-CoV-2 infected with HBV.
- What do these findings mean? (Separate paragraph)
Response: We would like to mention that this information is mentioned in the Conclusion section of the revised manuscript, page 20, lines 508–511. And presented below
Both SARS-CoV-2 and HBV infections were found to be highly prevalent in our leukemia patients’ cohort, suggesting the need for viral screening in hematological malignant cases to monitor infections and improve prognosis. Future studies with larger sample sizes may aid in delineating the patient outcomes of viral coinfections among leukemia patients.
- What is the impact of this research on society
è These points added in the Introduction section on lines 102-105, page 4 and on lines 119-121., page 5
- Key highlights/ Future direction must also be part of the manuscript. (Separate paragraph)
Response: - We agreed with the reviewer's comment. This recommendation
” Although the present study was the first on leukemia cases, highlighting the importance of laboratory screening for the presence of viral pathogens, future studies on the incidence of viral pathogens among leukemia patients at a large sample size is needed’’
is added in the revised manuscript. On lines 122-124, page 5
- Table 1, Table 2., some typing errors make it difficult to understand it.
It very difficult to find the missing words, please correct the table and resubmit for review.
Response: Comments were accepted, and corrections were done accordingly.
We inserted/corrected word/s in Table 1(line 280 and Table 2(line 377) -
Thank you for your valuable comments
Point by point response for Reviewer I
We would appreciate your constructive comments.
We presented our response by highlighting it with a yellow color in the main document.
- In my opinion, some recent references for the year 2023-24 are highly recommended. Therefore, I suggest the authors focus their efforts on researching the most recent and relevant literature: I believe that adding a few more studies will help to provide better and more accurate backgrounds this study. Please on the link (https://scholar.google. com/scholar? as_q= Hepatitis +B&as_epq=SARS+CoV+2&as_oq=&as_eq=&as_occt=title&as_sauthors=&as_publication=&as_ylo=2023&as_yhi=&hl=en&as_sdt=0%2C5&as_vis=1
Response: We thank the reviewer for the valuable suggestions. In response, in the Introduction section, paragraph 2, we added references 8, 9, 10, 12 and 13, recent papers that you suggested us to add). — see Page 4, from line 93-101.
- incorporate comprehensive flow sheet abstract (colored Scheme), which must be part of the manuscript, including all information that is ranging from material experiment parts, results and other key points mentioned in the text. This part is very important, as the reader will understand the whole manuscript without going into detail. See the following for example: https: //doi.org /10.1080/21655979.2020.1865607
Response: We thank the reviewer for raising this important point. In response, we produced and inserted an abstract along with a flow chart. —see Line 56 of Page 2 of the revised manuscript
- One page: Similarly, infographic abstract should also part of the manuscript
Response: We thank you for such valuable suggestions that strengthened the quality of our manuscript. In response, we produced and inserted an infographic abstract. – see line 76 of page 3 of the revised manuscript.
- Why was this study done?
Response: Thank you for this comment. It was overlooked in the first submission and thanks to you for highlighting now included in the introduction section, paragraph 3 of lines 102-to 105 of the revised manuscript.
- What did the author and co-authors do and find? (Separate paragraph)
Response: This is to kindly inform you that this information is stated in the Materials and Methods section on pages 5, 6, 7, and in the results on pages 8–15. However, we tried to present it in one paragraph in concise form, as presented below.
A cross-sectional study was conducted from July 2020 to June 2021 among acute leukemia cases. A standardized data collection form was used to collect demographic and clinical data, and 3 ml of blood samples were collected. Plasma samples were tested with ELISA for the presence of anti-SARS-CoV-2, HBsAg, anti-HIV, and anti-HCV, and PCR was used for HBV-DNA. The collected data were analyzed using SPSS version 25 statistical software, and the association between variables was analyzed by chi-square. A P-value of <0.05 was considered statistically significant. The SARS-CoV-2 seroprevalence was 35.5%. There was a statistically significant difference in the seropositivity of samples collected at different time periods. SARS-CoV-2 seropositivity was reported in the majority of regions of Ethiopia, and the prevalence of SARS-CoV-2 seropositivity was consistent with previous studies conducted in Ethiopia. In addition, 22.7% of the cases were positive for HBV, and 28% of SARS-CoV-2-seropositive cases were found to be positive for HBV infection. However, there were no statistically significant associations between SARS-CoV-2 seropositivity and HBV, HCV, or HIV seropositivity, or with sex, age category, Ethiopian region of residence, or acute leukemia subtypes. There was no difference in the liver function test or ALT between SARS-CoV-2 seropositive and HBV-infected cases and cases with seronegative SARS-CoV-2 infected with HBV.
- What do these findings mean? (Separate paragraph)
Response: We would like to mention that this information is mentioned in the Conclusion section of the revised manuscript, page 20, lines 508–511. And presented below
Both SARS-CoV-2 and HBV infections were found to be highly prevalent in our leukemia patients’ cohort, suggesting the need for viral screening in hematological malignant cases to monitor infections and improve prognosis. Future studies with larger sample sizes may aid in delineating the patient outcomes of viral coinfections among leukemia patients.
- What is the impact of this research on society
è These points added in the Introduction section on lines 102-105, page 4 and on lines 119-121., page 5
- Key highlights/ Future direction must also be part of the manuscript. (Separate paragraph)
Response: - We agreed with the reviewer's comment. This recommendation
” Although the present study was the first on leukemia cases, highlighting the importance of laboratory screening for the presence of viral pathogens, future studies on the incidence of viral pathogens among leukemia patients at a large sample size is needed’’
is added in the revised manuscript. On lines 122-124, page 5
- Table 1, Table 2., some typing errors make it difficult to understand it.
It very difficult to find the missing words, please correct the table and resubmit for review.
Response: Comments were accepted, and corrections were done accordingly.
We inserted/corrected word/s in Table 1(line 280 and Table 2(line 377) -

Reviewer 2 Report
Comments and Suggestions for Authors
Very interesting study also regarding epideniology in patients with leukemia in a geographical area with sometimes incomplete epimyological data. It would be very interesting to evaluate the phase of the disease in which positivity for Sars Cov 2 occurred in order to evaluate whether serological responses occurred in patients undergoing chemotherapy or not. The data about the treatment phase (at least for large groups) should be entered as well as if there are pre- and post-infection lymphocyte typifications.
Author Response
Thank you very much for your comments
Please find the attached documents
- point by point respnse
- Revised manuscript
Response to Reviewer II
We appreciate your valuable comments.
We presented our response by highlighting it with a yellow color in the main document.
Very interesting study also regarding epidemiology in patients with leukemia in a geographical area with sometimes incomplete epidemiological data. It would be very interesting to evaluate the phase of the disease in which positivity for SARS-CoV-2 occurred in order to evaluate whether serological responses occurred in patients undergoing chemotherapy or not. The data about the treatment phase (at least for large groups) should be entered as well as if there are pre- and post-infection lymphocyte typifications
Response : We thank the reviewer for raising these important points.
As stated in the Materials and Methods section on lines 138–139, page 6, all samples from all patients were obtained prior to chemotherapy. We did not assess lymphocyte counts in the study, as these can be quite unreliable in the setting of leukemia. (The diagnostic light scattering properties of leukemia cells sometimes overlap with those of lymphocytes.) Furthermore, since PCR tests for COVID-19 were not performed during sample collection (at the time of admission), we have no information regarding whether or not patients had active disease.

Reviewer 3 Report
Comments and Suggestions for Authors
The present manuscript reports about the seroprevalence of SARS-CoV-2 and Hepatitis B coinfection among immunocompromised individuals with haematological cancers.
Following are the comments after reviewing the manuscript:
1. There is no mention of vaccination status concerning HBV among the study population. Kindly add the same and comment if it had any effect on the data generated.
2. There is no mention of active SARS-CoV-2 infection among the study population.
3. The authors haven’t mentioned the seroprevalence determined in the study concerning which wave of COVID corresponds to which strain and does it has any effect on the overall prevalence among the study population.
4. The sample size is limited to draw out the proper conclusions in the present article.
5. Validation of the data needs to be done for non-cancer patients with SARS-CoV-2 infection. Authors are requested to kindly comment on the same.
6. Section 2.3 – Were plasma samples screened for IgM antibodies as well?
7. What is the reason for including 66 patients with acute leukemia before the initial COVID-19 infection? Kindly clarify.
8. As the present manuscript deals with seroprevalence among cancer patients, was it necessary to compare the data with different study groups like healthcare staff and young students? This doesn’t give any significant insight into the seroprevalence status.
9. Section 3.4 – Is the median value of 15 IU/ml (IQR-, 14-44) correct? Kindly check.
10. What was the level of ALT status and other liver enzyme status concerning SARS-CoV-2 and HBV coinfection? Kindly present the data.
11. Data from seroprevalence of coinfection in different types of leukemia is not available. Kindly present.
Author Response
Thank you very much for your comments
please find the attached document
Point by point response for reviewer III
We appreciated your exhaustive, valuable, and constructive comments.
We presented our response by highlighting it with a yellow color in the main document.
- There is no mention of vaccination status concerning HBV among the study population. Kindly add the same and comment if it had any effect on the data generated.
Response: We have no data on the HBV vaccination; however, we can reasonably assume that most, if not all, were HBV vaccine naïve since the vaccine was introduced to Ethiopia in 2007, meaning that at the initiation of the study (July 2020), only patients 13 or 14 years old would have been vaccinated, and reviewing our data indicated that only 3 cases were enrolled at that age and hence could have been vaccinated. This would not likely have substantially affected the overall results.
- There is no mention of active SARS-CoV-2 infection among the study population.
Response: -This was stated in the original document (in the Limitation section, page 17); Moreover, we have now stated this point in the revised version (line 496-497, page 19).
- The authors haven’t mentioned the seroprevalence determined in the study concerning which wave of COVID corresponds to which strain and does it has any effect on the overall prevalence among the study population.
Response: - We thank the reviewer for raising this important point.
This is now further explored in the Discussion section of the revised version (line 459-462, page 18)
- The sample size is limited to draw out the proper conclusions in the present article.
Response: -We were limited in the sample size because the hospital was primarily focused on managing the pandemic, so the number of leukemia cases that visited was restricted. Since, various departments, including hematology sections, switched their responsibilities to the handling of COVID-19 cases; therefore, including a large sample size was very challenging since this hospital is the only one in Ethiopia to manage cancer cases. Even though the sample size was limited, the study participants were from different regions of the country; furthermore, this was the first study on leukemia cases in Ethiopia, and we recommend further studies with a large sample size. We acknowledged these limitations (line 498-501, page 19).
- Validation of the data needs to be done for non-cancer patients with SARS-CoV-2 infection. Authors are requested to kindly comment on the same.
Response: -Thank you for this comment. It is true that our study was limited by the lack of a direct comparison between leukemia patients with other non-leukemia patients, otherwise matched, but these would be very difficult to do, in particular considering the diverse regions of the country the leukemia patients were coming from. Most of the hospital deals with patients living closer to the capital city. It is for these reasons that we summarized our data in the context of other studies done in the country, some of which were done in a similar time frame and covered multiple regions of the country. We have added similar statements to the limitations section line of the revised version (lines 501-506, page 19). We would argue the most relevant study to be included was that done by our same institute at approximately the same time using the same in-house assay. This point has now been added in the Discussion section (lines 449-451, page 18) for clarity.
- Section 2.3 – Were plasma samples screened for IgM antibodies as well?
Thank you for your good question raised
Response: No, they weren’t screened specifically for IgM, but all IgG positive plasma were confirmed to be IgG, IgM and IgA positive by an alternate commercial test. We now add new text clarifying this in the materials and methods in section 2.3 (lines 154—155 page 6), as well as limitations section (lines 497-498, page 19)
- What is the reason for including 66 patients with acute leukemia before the initial COVID-19 infection? Kindly clarify.
Response: -As stated, under section 3.2 (lines 303- 307 page 11), this was a negative control cohort for the IgG assay we used for serodetection. We have modified the text in the materials and methods, to emphasize this (lines 144-147 page 6).
- As the present manuscript deals with seroprevalence among cancer patients, was it necessary to compare the data with different study groups like healthcare staff and young students? This doesn’t give any significant insight into the seroprevalence status.
Response :- The point of this comparison was just to see if seroprevalence of SARS in leukemia patients matched or not that in appropriate regions of the country, adjusted for time. How seropositivity might have been impacted by presence of leukemia, in particular immunocompromise, is in the discussion ( lines 449-451, page 18).
- Section 3.4 – Is the median value of 15 IU/ml (IQR-, 14-44) correct? Kindly check.
Response: - Thank you very much for your concern. We have checked the result (recalculated) and confirmed it as stated. There were cases with much higher values than the median value .
- What was the level of ALT status and other liver enzyme status concerning SARS-CoV-2 and HBV coinfection? Kindly present the data.
Response: - We did ALT tests for the cases, but due to a shortage of reagents, we did not measure other liver enzyme tests such as AST. Per your suggestion, we presented the ALT data on HBV infected SARS seropositive or seronegative in a new Table 4 (at the end of the Result section) line 408, page 16
- Data from seroprevalence of coinfection in different types of leukemia is not available. Kindly present.
Response: -This is now presented in a new Table 3(line 390 , page 16). and associated text in the Results section (line 386-388-page 15).

Reviewer 4 Report
Comments and Suggestions for Authors
The article concerns serological markers of SARS-CoV-2, HIV, HBV and HCV infections (ELISA) and HBV DNA (PCR) among acute leukemia patients in Ethiopia at diagnosis before anticancer treatment. The study included 110 patients at a single clinical center. It showed clear dependence of SARS seropositivity on the period of pandemics. No correlations with other tested infections or different clinical variables were revealed. The results for SARS-CoV-2 are compatible with other international studies on the item.
Remarks and questions:
Materials and methods:
Line 73: did the 66 previously admitted patients represent a separate comparison group for the SARS group? Are there any data on HIV, HBV and HCV in this group?
Line 95: Please specify if the Micro ELISA-HIV does detect HIV-1 vs HIV-2 (or both) and discriminate between them?
Results:
Line 206: … HBsAg negative but HBV DNA positive…If these samples were tested from the same samples, or at different periods of time These discrepancies may be caused by additional transfusions during therapy of leukemia.
In Discussion, one should mention the SARS-CoV-2 strains detected in Ethiopia during the period of study (2020-2021) referring, e.g., to the WHO data.
In conclusion: In potential limitations of the study, one should indicate the single-center investigations, thus suggesting specific regional features of the SARS-Cov-2 epidemiology over the period of study (2020-2021).
There are some misprints noted, e.g.: line 128: …distribution of leukemia… - should be …leukemia cases…
Comments on the Quality of English LanguageMinor editing of English is recommended
Author Response
Thank you very much for your constructuve comments
please find the attached document
Point by point response for Reviewer 4
We would like to acknowledge your valuable and constructive comments
We presented our response by highlighting it with a yellow color in the main document.
Line 73- Line 73: did the 66 previously admitted patients represent a separate comparison group for the SARS group? Are there any data on HIV, HBV and HCV in this group?
Response: -Yes, the 66 cases were separate group samples collected before the report of COVID-19 in Ethiopia and we intended to use primarily is a negative control group to validate the SARS-CoV-2 serological assay. However, we do have data on HIV, HBV, and HCV for these 66 cases, and this is now presented in the results text, at 3.4 (lines 359-362 page 14).
Line- 95 – ELISA KIT FOR HIV
Line 95: Please specify if the Micro ELISA-HIV does detect HIV-1 vs HIV-2 (or both) and discriminate between them?
Response: - We used Micro ELISA–HIV Ag & Ab, J. Mitra & Co. Pvt. Ltd., New Delhi, India to detect HIV-1 and/or HIV-2 and HIV-1 p24 antigen. This is now clarified in the materials and methods. Section 2.5 (lines 175-177, page 7)
Results
Line 206: … HBsAg negative but HBV DNA positive…If these samples were tested from the same samples, or at different periods of time. These discrepancies may be caused by additional transfusions during therapy of leukemia.
Response: -All samples were collected at one time only, i.e. patient recruitment, so no follow up samples were obtained; this is stated in the materials and methods. Sections 2.2 of the revised version (line 143-144, page 6)
In Discussion,
One should mention the SARS-CoV-2 strains detected in Ethiopia during the period of study (2020-2021) referring, e.g., to the WHO data.
Response -Thank you very much for your comment
This is now included in the discussion part of the revised version. lines 459-462.

Round 2
Reviewer 1 Report
Comments and Suggestions for Authors
Thanks for replying the comments
Author Response
Dear, reviewer,
We would like to acknowledge your continued assistance to improve the document and appreciate your constructive comments. This is to kindly inform you that the updated version now emphasizes the following key points:
- Version with edits: One of the co-authors, a native English speaker, has edited the manuscript.
- As per the Editor suggestion, we removed the abstract along with the flow sheet; hence, the amended version only has the graphic abstract.
- A few adjustments and the modified points are highlighted in yellow.
- We removed a few references from the document without jeopardizing its integrity.
Kindly refer to the revised version in the attachment
We would appreciate hearing from you if you have any questions, suggestions, or comments.
Thank you again for your usual support!
Best
Jemal Alemu, corresponding author

Reviewer 3 Report
Comments and Suggestions for Authors
Thank you for providing clarifications regarding the raised concerns.
Comments on the Quality of English LanguageThe grammatical mistakes and typos need to be thoroughly checked and rectified.
Author Response
Dear, reviewer,
We would like to acknowledge your continued assistance to improve the document and appreciate your constructive comments. This is to kindly inform you that the updated version now emphasizes the following key points:
- Version with edits: One of the co-authors, a native English speaker, has edited the manuscript.
- As per the Editor suggestion, we removed the abstract along with the flow sheet; hence, the amended version only has the graphic abstract.
- A few adjustments and the modified points are highlighted in yellow.
- We removed a few references from the document without jeopardizing its integrity.
Kindly refer to the revised version in the attachment.
We would appreciate hearing from you if you have any questions, suggestions, or comments.
Thank you again for your usual support!
Best
Jemal Alemu, corresponding author
